# cyjShiny: A cytoscape.js R Shiny Widget for network visualization and analysis

**Augustin Luna** [1,2]*, **Omar Shah**[3], **Chris Sander**[1,2], **Paul Shannon**[4]

**1** Department of Systems Biology, Harvard Medical School, Boston, MA, United States of America, **2** Broad Institute of Harvard and MIT, Cambridge, MA, United States of America, **3** Red Kubes, Utrecht, Netherlands, **4** Institute for Systems Biology, Seattle, Washington, United States of America

* cannin+plos@gmail.com

**Data Availability Statement:** The package is available from the CRAN R repository: cran.r-project.org/web/packages/cyjShiny/index.html. Documentation and source code for cyjShiny is

## Abstract

cyjShiny is an open-source R package that allows users to embed network visualization into Shiny apps and R Markdown documents. cyjShiny (https://github.com/cytoscape/cyjShiny) builds on the cytoscape.js Javascript graph library. Additionally, the package provides helper functions to convert common R data representations (e.g., data.frame) into forms compatible with cytoscape.js.

## Introduction

Interactions in various forms (e.g., regulatory, metabolic, neuronal, ecological interaction networks) are fundamental to our understanding of biological phenomena. Further, researchers are increasingly consuming scientific information through web-based data portals. R Shiny (shiny.rstudio.com) is a framework that simplifies the development of interactive web applications for research projects using the R programming language. The Shiny framework for web development is extensible using the htmlwidgets R library (htmlwidgets.org) that allows the use of Javascript-based visualization libraries in R. Here we present the cyjShiny R package, a wrapper package for the cytoscape.js. cytoscape.js. is a widely used Javascript library for network analysis and visualization that is extensible through plugins [1].

The use case for cyjShiny is differentiated from either cytoscape.js and Cytoscape Desktop [1,2]. In comparison to cytoscape.js, cyjShiny simplifies the effort needed (by users who may be familiar with R but unfamiliar with Javascript) to embed interactive network visualizations into either R Markdown-based reports or R Shiny-based web applications. While Cytoscape Desktop is a ready-to-use platform for end-users to visualize and analyze their data, cyjShiny is a building block for developers to integrate into their existing databases, tools, or workflows; see the Use Case section for examples.

## Methods

### Implementation

cyjShiny is implemented in the R programming language; the R package is compatible with versions of R 3.5 and above.

*Data Formats*: cytoscape.js requires a JSON-based input data structure [1]. To facilitate the creation of this data structure, cyjShiny provides two helper functions graphNELtoJSON() and

available at github.com/cytoscape/cyjShiny under the MIT license.

**Funding:** The authors received funding from the Google Summer of Code program and funding for the National Resource for Network Biology (NRNB) from the National Institute of General Medical Sciences (NIGMS P41 GM103504). The funders had no role in study design, data collection and analysis, decision to publish, or preparation of the manuscript.

**Competing interests:** The authors have declared that no competing interests exist.

dataFramesToJSON(). First, the graphNELtoJSON() function allows users to convert network data that they may have in the graphNEL format from the graph R package. Users of the igraph R package can convert to the graphNEL representation using the as_graphnel() method found in igraph. Second, the dataFramesToJSON() function can convert a data.frame to the necessary JSON representation. This function takes the parameter, tbl.edges, which should have three columns source, edge, and interaction; source and target are identifiers for nodes, while interaction provides a category for the interaction in the form of a string. If users wish to include node attributes (e.g., values or categories to be used for node styling), users may provide a second optional data frame: tbl.nodes. The tbl.nodes data.frame must include an "id" column to be mapped to the source or target columns of the tbl.edges data.frame. The result of either of these functions is then input to the cyjShiny() method which will render the network.

*Network Styling*: Additionally, to the network data, users must provide styling information. cytoscape.js uses a styling format heavily inspired by Cascading Style Sheets (CSS) to provide this functionality. All of a user's style choices, simple to complex, must be encoded into a style file, the path to which may be passed to cyjShiny when an instance is constructed, or subsequently in an R function call, *loadStyleFile("yourStyle.js")*, from within the Shiny application. This means that users can, for instance, set the node property "border-color" to "black" to have nodes with black borders consistent with how one would style a webpage table. More information regarding cytoscape.js styling is available here: js.cytoscape.org/#style.

*Plugins*: Cytoscape.js provides an application programming interface (API) that allows the development of plugins to extend its functionality. These plugins provide functionality not present in the core library. Currently, there are over 60 user-contributed extensions (i.e., plugins) for cytoscape.js (js.cytoscape.org/#extensions). cyjShiny pre-packages several plugins to provide various network layouts. For example, two of the included algorithms are fCoSE (Fast Compound Spring Embedder) layout and the Klay algorithms [3]. These two algorithms are suitable for compound networks; a biological example of a compound network is a network that includes protein complexes and there is interest in keeping predefined complexes as a unit.

## Use cases

*Shiny Application*: We provide several demos as part of the package (github.com/cytoscape/cyjShiny/tree/master/inst/demos). Fig 1 shows the demo (github.com/cytoscape/cyjShiny/blob/master/inst/demos/basicDemo/app.R) using a sample yeast protein-protein interaction network. This demo covers many key features that users are likely to want. Including basic requirements of showing the network within a Shiny app, but also, more advanced features such as 1) switching between different data (to color nodes under different conditions) and stylings on the same network, 2) highlighting nodes, 3) fitting the network to the available screen space, and 4) animating networks by looping through dataset visualized on the network. The demo is available online at https://cyjshiny.shinyapps.io/basicDemo. The project website README provides direct links to various demos including a minimal quick start example to more complex demos that 1) show how a network can utilize pre-generated layouts and 2) utilize networks generated via Cytoscape Desktop, as well as others; links on the README will be updated as demos are added or removed. We further provide documentation to explain how networks on cyjShiny can be styled to suit developer needs.

*R Markdown*: While cyjShiny was developed for inclusion in Shiny applications, cyjShiny can be embedded in R Markdown-based scientific notebooks. Such notebooks provide a report for a given analysis that presents code alongside its outputs (e.g., plots, tables), as well as prose text explaining the code and output. An example notebook has also been provided in the source code that embeds a network visualization widget.

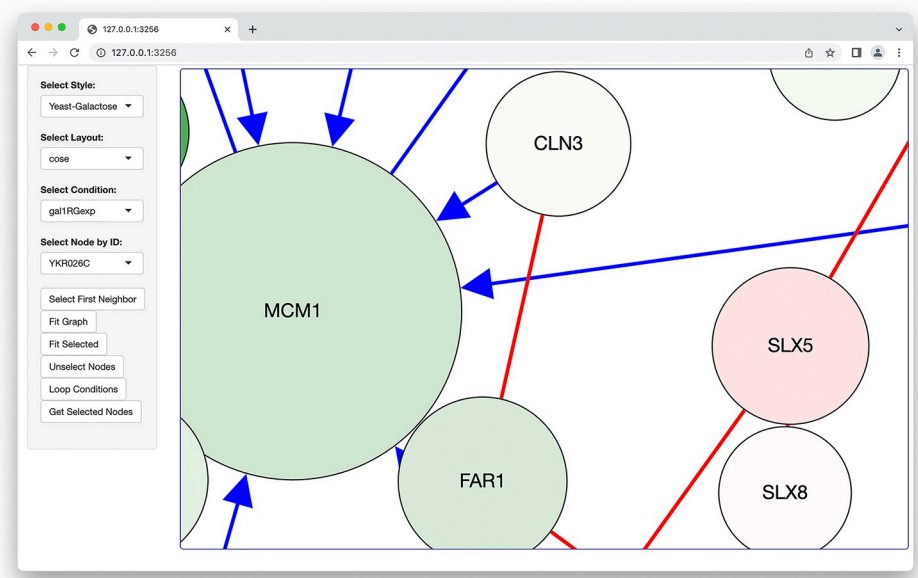

**Fig 1. Screenshot of a demo Shiny application with a sample yeast protein-protein interaction network.** Menus shown on the left-hand sidebar allow users to control various properties of the network including the layout and the data overlaid on nodes.

*Examples*: Many biological databases utilize cytoscape.js for network visualization of molecular interactions and co-expression networks across a wide range of topics, including FlyBase, FerrDb, IntAct, MolluscDB, DEPOD, ATTED-II, and Pathway Commons [4–10]. Likewise, developers of gene set analysis methodologies, such as g:Profiler and WebGestalt, have utilized cytoscape.js to visualize both interaction networks as well as Gene Ontology hierarchies [11,12]. Using such databases and tools researchers can interactively explore datasets and results via web applications [13,14]. By utilizing cyjShiny developers can both develop methodologies and web applications utilizing only R code thereby lowering the barrier to creating such web-based interactive tools. This strategy has been employed by the developers of Bayes-NetBP that utilize R Shiny for the exploration of results to their method analyzing Bayesian networks [15].

## Conclusion

cyjShiny facilitates the integration of interactive network visualizations into Shiny apps and R Markdown documents using only the R programming language. A tutorial (vignette) and demo applications have been provided as part of the package for users to learn the customization options available. These interactive visualizations enrich the exploration of user datasets and can be customized to address the needs of specific projects.

## Acknowledgments

We thank the developers of cytoscape.js and the maintainers of the R CRAN repository. Additionally, we would like to thank the coordinators of the Google Summer of Code program.

## Author Contributions

**Conceptualization:** Omar Shah, Chris Sander, Paul Shannon.

**Funding acquisition:** Chris Sander.

**Software:** Augustin Luna, Omar Shah, Paul Shannon.

**Supervision:** Paul Shannon.

**Writing – original draft:** Augustin Luna, Chris Sander, Paul Shannon.

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
