## [Decision Letter · Decision Letter 0]

23 Feb 2023

PONE-D-22-24150

cyjShiny: A cytoscape.js R Shiny Widget for Network Visualization and Analysis

PLOS ONE

Dear Dr. Luna,

Thank you for submitting your manuscript to PLOS ONE. After careful consideration, we feel that it has merit but does not fully meet PLOS ONE’s publication criteria as it currently stands. Therefore, we invite you to submit a revised version of the manuscript that addresses the points raised during the review process.

We look forward to receiving your revised manuscript.

Kind regards,

Yanbin Yin

Academic Editor

PLOS ONE

Journal Requirements:

4. Please ensure that you include a title page within your main document. You should list all authors and all affiliations as per our author instructions and clearly indicate the corresponding author.

Additional Editor Comments:

Sorry for being late. Reviews came in delayed due to the winter break. Please consider reviewer #4's comment to include detailed tutorial to make the tool more user friendly.

Reviewers' comments:

Reviewer's Responses to Questions

**Comments to the Author**

1. Is the manuscript technically sound, and do the data support the conclusions?

Reviewer #1: Yes

Reviewer #2: Yes

Reviewer #3: Yes

Reviewer #4: Partly

2. Has the statistical analysis been performed appropriately and rigorously? 

Reviewer #1: N/A

Reviewer #2: N/A

Reviewer #3: N/A

Reviewer #4: N/A

3. Have the authors made all data underlying the findings in their manuscript fully available?

Reviewer #1: Yes

Reviewer #2: Yes

Reviewer #3: Yes

Reviewer #4: Yes

4. Is the manuscript presented in an intelligible fashion and written in standard English?

Reviewer #1: Yes

Reviewer #2: Yes

Reviewer #3: Yes

Reviewer #4: Yes

5. Review Comments to the Author

Reviewer #1: The authors present cyjShiny, a tool in R for visualising networks with Shiny. cyjShiny provides a mechanism for interactively exploring network data in a web browser, as generated from analyses (e.g. bioinfo.) in R. This software package makes exploration and demonstration of network visualisations accessible to researchers who are familiar with R notebooks -- but who may not be so familiar with other programming environments, such as web technologies.

The use of cyjShiny promises to ease the interpretation and dissemination of network data in research, and so it is my opinion that this article is suitable for publication.

I have no suggestions re. revisions.

Reviewer #2: The manuscript is well-written. The R package is easy to install. The authors have also provided several demos on their GitHub page with the complete codes available. My only concern is that the current R package ShinyApp) is not very user-friendly to users with no or limited R knowledge. I would suggest adding more details to the user manual and allowing the user to upload and visualize the data immediately after launching the ShinyApp without writing too much code.

Reviewer #3: The authors propose a web portal called cyjShiny for visualizing network data. This application is built using cytoscape.js and provides APIs for converting graph data from dataframe format in R or iGraph format into JSON format. While the authors do not present many novel innovations in cytoscape.js, this tool is still useful for researchers who use R to write Rmarkdown or develop websites.

There are a few areas that need improvement:

1. The README.md and WIKI in the GitHub repository should be carefully rewritten. For example, in https://github.com/cytoscape/cyjShiny/wiki/installation, the script tinyApp.R is mentioned, but it is not present in the repository.

2. It would be helpful if an online portal was provided to showcase demos of the tool.

Reviewer #4: The authors presented an R package cyjShiny that allows users to embed network visualization into Shiny apps and R Markdown documents. I installed the R package on Windows 10 and tested the several demos provided by the authors. The demos and functionalities work smoothly. My concern is that the help tutorials and the manuscript is too short and not user-friendly at all. The authors should spend more words and paragraphs to improve the manuscript. For example, how can the R package cyjShiny be used in scientific studies? What’s the relationship between cyjShiny, and other tools including cytoscape.js and cytoscape? The help manual for cyjShiny is too simple. I can not find a helpful tutorial on how to use this R package. If you do not dig into the code, you will not be aware the existence of 10 demos provided by the authors.

6. PLOS authors have the option to publish the peer review history of their article (what does this mean?). If published, this will include your full peer review and any attached files.

Reviewer #1: No

Reviewer #2: No

Reviewer #3: No

Reviewer #4: **Yes: **Wen Yao

---

## [Author Response · Author response to Decision Letter 0]

3 Apr 2023

The included resubmission cover letter describes specific changes to the text in response to reviewer feedback.

---

## [Decision Letter · Decision Letter 1]

20 Apr 2023

cyjShiny: A cytoscape.js R Shiny Widget for Network Visualization and Analysis

PONE-D-22-24150R1

Dear Dr. Luna,

We’re pleased to inform you that your manuscript has been judged scientifically suitable for publication and will be formally accepted for publication once it meets all outstanding technical requirements.

Kind regards,

Yanbin Yin

Academic Editor

PLOS ONE

Additional Editor Comments (optional):

Reviewers' comments:

Reviewer's Responses to Questions

**Comments to the Author**

1. If the authors have adequately addressed your comments raised in a previous round of review and you feel that this manuscript is now acceptable for publication, you may indicate that here to bypass the “Comments to the Author” section, enter your conflict of interest statement in the “Confidential to Editor” section, and submit your "Accept" recommendation.

Reviewer #3: All comments have been addressed

Reviewer #4: All comments have been addressed

2. Is the manuscript technically sound, and do the data support the conclusions?

Reviewer #3: Yes

Reviewer #4: Yes

3. Has the statistical analysis been performed appropriately and rigorously? 

Reviewer #3: No

Reviewer #4: N/A

4. Have the authors made all data underlying the findings in their manuscript fully available?

Reviewer #3: Yes

Reviewer #4: Yes

5. Is the manuscript presented in an intelligible fashion and written in standard English?

Reviewer #3: Yes

Reviewer #4: Yes

6. Review Comments to the Author

Reviewer #3: (No Response)

Reviewer #4: (No Response)

7. PLOS authors have the option to publish the peer review history of their article (what does this mean?). If published, this will include your full peer review and any attached files.

Reviewer #3: No

Reviewer #4: **Yes: **Wen Yao

---

## [Editor Report · Acceptance letter]

3 May 2023

PONE-D-22-24150R1 

cyjShiny: A cytoscape.js R Shiny Widget for Network Visualization and Analysis 

Dear Dr. Luna:

I'm pleased to inform you that your manuscript has been deemed suitable for publication in PLOS ONE. Congratulations! Your manuscript is now with our production department. 

Kind regards, 

on behalf of

Dr. Yanbin Yin 

Academic Editor

PLOS ONE